# A case control study to investigate differences in motor control between individuals with and without non-specific low back pain during standing

Cathrin Koch[1☯], Augusto Garcia-Augundez[2‡], Stefan Göbel[2‡], Frank Hänsel[1☯]*

**1** Institute of Sport Science, Technical University Darmstadt, Darmstadt, Germany, **2** Multimedia Communications Lab, Technical University Darmstadt, Darmstadt, Germany

☯ These authors contributed equally to this work.
‡ These authors also contributed equally to this work.
* haensel@sport.tu-darmstadt.de

**Data Availability Statement:** The data underlying the results presented in the study are available from the Center for Open Science under the following link https://osf.io/4fwc9/.

## Abstract

Recent literature has indicated altered motor control in individuals with non-specific low back pain (NSLBP). These individuals present variations in back muscular activity and center of mass (CoM) oscillations. The aim of this study is to explore the possibility of quantitatively measuring these differences using standard parameters with electronic devices. Twenty individuals with NSLBP and 20 healthy controls, matched by sex and age, performed a total of three trials under three different conditions for 90 seconds each. These conditions were standing on firm ground with eyes open, with eyes closed and standing on unstable foam with eyes open. Balance data was acquired via a Kistler force platform and muscular activity was measured by electromyography derived bilaterally from the erector spinae. Afterwards, participants were asked to complete a questionnaire on their current mood, pain rating, well-being, disability and physical activity. Descriptive data from the questionnaire showed an average acute pain score of 2.6 and an average pain score of 5.1 for the prior six weeks in the NSLBP group, while the control group reported an acute pain of 0.1 and an average pain of 0.5. For wellbeing, differences were found only for the physical scale. Average disability was low for the NSLBP group. No differences in physical activity were found among groups. A repeated measures ANOVA did not show significant differences between groups for any parameter. There was also no main effect for the standing conditions and no interaction between group and condition. Simultaneously measuring biomechanical and neuromuscular parameters, allowed for a fine grain approach to understanding motor control in individuals with NSLBP. This study did not find differences as described in the literature, and suggests further examination of factors involved in pain and control processes to better understand implications of NSLBP and possible avenues for support.

**Funding:** - C.K. & A.G.-A. were funded by the Klaus Tschira foundation (Gemeinnützige GmbH), Germany - grant number: 00.280.2015 - https:// www.klaus-tschira-stiftung.de/ - The funders had no role in study design, data collection and analysis, decision to publish, or preparation of the manuscript.

**Competing interests:** The authors have declared that no competing interests exist.

## Introduction

The lifetime prevalence of low back pain in industrial countries has risen to 84% [1]. Approximately 85% of back pain is non-specific, which means that no structural change, inflammation or specific disease can be found as the cause of pain [2]. The number of people requiring treatment is high, however the reasons for non-specific back pain are unclear [3]. Recent studies have shown that many individuals tend to develop low back discomfort, as a precursor to non-specific low back pain (NSLBP), during periods of standing [4, 5]. Differences in trunk muscle activity [6] and center of pressure (CoP) displacement [5] precede this discomfort. It is assumed that these differences reveal altered motor control.

Models provide differing theoretical foundations for the relation between NSLBP and altered motor control [7, 8]. Hodges and Tucker [7] propose that pain, injury, or threat of pain and injury can lead to altered muscular activity and movement patterns to protect structures from further stress. This activity is redistributed within and between muscles which changes mechanical behavior and creates stiffness [7]. However, there is no stereotypical change in muscle activity for individuals with low back pain [7]. Examining specific tasks and other individual differences like the exact pain region, might indicate differences in muscle activity redistribution for people with NSLBP.

In the case of standing, recent reviews have confirmed altered motor control in NSLBP [9–11] based on different parameters. Two of these reviews [9, 10] investigate changes in CoP data during different quiet standing conditions. While the third [11] takes CoP, lumbo-pelvic angles as well as neuromuscular parameters into consideration. They found differences in motor control. However, the results of the reviews are inconsistent. For example, regarding CoP data, results of the reviews differ depending on the test condition. For quiet standing on a firm surface two reviews [9, 11] do not find evidence for differences between individuals with and without NSLBP, while the other one [10] reports differences between groups. For more challenging conditions like standing with closed eyes or standing on foam results are also different. Two reviews [10, 11] found differences between individuals with and without NSLBP and one [9] could not find differences Thus, a further replication study, which investigates the biomechanical and neuromuscular parameters in the same sample simultaneously and under different conditions is necessary. To our knowledge there is no study that synchronously records biomechanical and neuromuscular data and considers motor control on both levels.

The aim of this study is to verify findings of back muscle activity and CoM sway. Based on the existing model of adaptation to pain and recent studies, we hypothesize that there are measurable differences in motor control between individuals with and without NSLBP during quiet standing.

## Materials and methods

This case control study has been preregistered on the Open Science Framework (OSF). It can be found at osf.io/2tfqh.

### Participants

A sample of 32 participants (16 per group) was required to provide 80% power for evaluating effect size equal to 0.65 for each variable based on repeated measures ANOVA with a significance level of 0.05. Aiming for this effect size, we recruited 20 participants per group to account for potential dropout. Twenty patients with a history NSLBP (11 females; 28 years +/-3.9; 174.3 cm +/-7.3cm; 70.9 kg +/-10.9kg) and 20 healthy adults (11 females; 26 years +/-4.1; 172.8 cm +/-8.4cm; 69.8kg +/-10.4kg) were recruited from the community through flyer postings, advertisements and word of mouth. The healthy adults were matched according

to age and sex. Participants were included if they had NSLBP for 6 months or more and excluded if they had serious spinal pathology, nerve root pain, previous history of spinal surgery, or structural deformity of the spine, as judged by an expert physical therapist. They had no history of pain extending beyond the gluteal fold. None of the participants had a history of uncorrected vision impairment, vestibular or respiratory disorders, auditory or cognitive deficits, diabetes mellitus, recent lower limb pathologies, pregnancy or use of any medicine that might affect balance. All participants gave informed consent, which was approved by the Ethics Committee of Technical University Darmstadt (TU Darmstadt).

The data of two participants regarding current mood could not be evaluated due to incomplete questionnaire reporting. Additionally, sports data for physical activity of three individuals per group was missing. Each individual received 15€ for participation. Finally, there were no significant differences in body height, body weight, and standing width between groups, all p´s ≤ .210.

## Procedure

The study took place at the sport science laboratory of the TU Darmstadt. After describing the informed consent with information about the experiment, participants could familiarize themselves with standing on the foam pad. Foot position for all trials was then determined. This position required that participants were standing in the middle of the force plate with their medial malleolus at the center with the same distance for the right and left foot. All trials were conducted barefoot. Participants were asked to stand naturally to mimic their normal balance. Measuring devices were attached to the individuals. Submaximal voluntary contractions (subMVC) were performed to normalize EMG data, following Dankaerts, O'Sullivan [12]. Participants had to stand as still and relaxed as possible under three different conditions. These conditions were (1) standing on a stable surface with eyes open (EO), (2) standing on a stable surface with vision occluded (EC), and (3) standing on foam with eyes open (FO). Each condition was performed three times for 90 seconds in a counterbalanced order. All trials were performed on a force plate in a quiet laboratory setting. Participants were instructed to stand on the force plate with their feet apart at the determined distance, to look straight ahead and keep their arms at their sides in a comfortable position.

Afterwards, a questionnaire was given to determine emotional stress during the standing procedure. It consisted of questions about current emotional pressure assessed with six items [13] to determine whether stress may have been a factor in their performance. Additionally, they completed a numeric rating scale [14] for current pain and pain over the last six weeks, a German version of the Roland and Morris Disability Questionnaire [15], questions about the general state of health (SF-12) [16], and a physical activity questionnaire [17].

## Equipment/Measurement systems

To measure CoM data and ground reaction forces a force platform was used (Kistler model 9287C8). The output of the force plate was sampled at 1000 Hz.

Force plate and surface electromyography (sEMG) data were collected simultaneously during standing. Muscle activity patterns of the participants were measured by using the multichannel bipolar sEMG biosignal amplifier USBAMP and data acquisition module GammaSys developed by G.tec (www.gtec.at). Bipolar sEMG was measured with solid gel electrodes, placed bilaterally with an inter-electrode distance of 23mm on the muscle belly after cleaning the patient's skin with abrasive paste. Electrodes were placed on the left and right side, lateral to the first lumbar processus spinosus (L1), according to the surface electromyography for the noninvasive assessment of muscles guidelines, and lateral to the fourth lumbar processus

spinosus (L4) (for a total of 4 channels). Lumbar vertebrae were located through palpation, using the iliac crest as a landmark. A reference electrode was placed over the right spinae iliac anterior superior. In the preprocessed method, sEMG data were collected with a sampling rate of 1024 Hz for standing. A band–pass-filter of 17 to 500 Hz was applied.

### Data processing and analysis

Data from the force plate was collected by using an own application developed for Microsoft Visual Studio based on the software provided by Kistler. This data was sent via UDP to a Simulink model developed using Matlab (www.matlab.com). We employed Simulink because G.tec provides a direct interface between the biosignal amplifier and this software, meaning all data could be acquired synchronously. Afterwards, data were filtered by a 5 Hz low pass filter.

From the acquired data, the following parameters were calculated: Root mean square (RMS) of CoM sway and average EMG activity for each trial. To allow comparison between individuals, all EMG data were normalized by the subMVC. The subMVC data were recorded while the subject was in the prone position with the knee flexed 90˚ on an examination table and then in the prone position with both legs lifted 5 cm off the table [12, 18]. The subMVC data were accepted by the primary investigator when subjects could lift their legs about 5 cm off the table and hold the position for at least 3 seconds. The highest value in this position was used to normalize the other sEMG data. After acquiring sEMG during the standing conditions, data were filtered by a band-pass filter between 17 and 500 Hz to eliminate heart rate from the signal.

Force plate and sEMG data were analyzed with a 2x3 MANOVA (Group (NSLBP, no NSLBP) x Condition (EO, EC, FO)), with group and condition as a repeated measurement factor, since groups were matched according to age and sex.

## Results

### Questionnaire

The significance threshold was set at .05 for all statistical tests. Evaluating the results of the questionnaire, current mood did not differ between groups ($t(17) = 1.301$, $p = .211$). Concerning pain, participants with non-specific low back pain rated their current pain intensity as 2.6 points (+/-1.7 SD) on average and their pain over the last six weeks as 5.1 points (+/-1.4 SD) on average with a 10-point visual analogue scale. The average value in the control group was 0.1 points (+/- 0.4 SD) for current pain and 0.5 points (+/- 0.7 SD) over the last six weeks. None of the healthy controls reported a pain intensity higher than three in the last six weeks, while in the NSLBP group no one reported less than three. Average disability in the Roland and Morris Disability Questionnaire was 4.2 points (+/-2.2 SD) for the NSLBP group, which is interpreted as low. The control group did not report any disability. Concerning the physical scale on the SF12, individuals with low back pain showed significantly lower values (NSLBP 48.0 points +/-8.0 SD vs. control 55.9 points +/-4.4 SD; $t(19) = -4.686$, $p = .001$), whereas there is no difference in the mental healthscale (NSLBP 48.0 points +/-7.1 SD vs. control 48.6 points +/-10.0SD; $t(19) = -0.085$, $p = .933$). The Physical Activity Questionnaire did not show any significant difference between groups (sports: NSLBP 4.0 +/-0.2 SD vs. control 3.8 +/-0.1 SD; $t(14) = 0.793$, $p = .441$; work: NSLBP 2.4 +/-0.5 SD vs. control 2.5 +/-0.4 SD; $t(19) = -0.763$, $p = .455$; free time: NSLBP 3.7 +/-0.2 vs. control 3.6 +/-0.2 SD; $t(19) = 0.001$, $p = .999$).

### Motor control

Data were analyzed with a 2x3 MANOVA (Group (NSLBP, no NSLBP) x Condition (EO, EC, FO)), with group and condition as a repeated measures factor. Hotellings'$T^2$, a test statistic of

the repeated measure MANOVA, showed a main effect for the different conditions ($F(4,72)$ = 5.77, $p < .001$, $\eta^2$ = .243) (see Table 1). However, there were no significant differences between groups ($F(2,18)$ = 0.51, $p = .606$, $\eta2$ = .054). In addition, there was no interaction effect for group ($F(4,72)$ = 0.20, $p = 937$, $\eta^2$ = .011). Table 2 shows the descriptive data for both groups.

Since we found a main effect for conditions, we performed two ANOVAs for the independent variables to clarify this effect. Mauchly´s test indicated a violation of sphericity. Therefore, we used the Greenhouse-Geisser correction to address the data. The results show a significant effect for CoM sway ($F(1.88,35.66)$ = 10.45, $p < .001$, $\eta^2$ = .355), but not for EMG data ($F(1,19.02)$ = 2.07, $p = .166$, $\eta^2$ = .098). Therefore, the significant main effect for condition is due to the CoM parameter. Post-hoc test with Bonferroni correction for CoM sway between different conditions demonstrate that standing on a foam pad is different from both standing on solid ground with eyes open ($p = .029$) and eyes closed ($p = .001$). Descriptive data for CoM sway is presented in Table 3.

## Discussion

We examined postural sway during quiet standing on firm and unstable surfaces with and without eyes closed. Contrary to our prediction, our results do not confirm that differences in motor control exist between individuals with and without NSLBP. We did not find any significant differences between groups in any of the parameters for CoM and EMG. However, at the descriptive level there is a difference between groups. The CoM values are higher for individuals with NSLBP (Table 2), which is consistent with the theory that CoM sway is higher due to impaired postural control. Examining the data more closely, the group difference may not have reached significance due to high interindividual variance of CoM sway. The effect size for CoM is also low ($\eta^2$ = .054). Among the eyes closed and unstable conditions, we found no significant interactions. This means that we also cannot confirm differences in more demanding quiet standing tasks.

However, there is a significant main effect for conditions. Standing on firm ground differs from standing on unstable ground for CoM sway, indicated by post-hoc tests. This suggests that the surface manipulation imposed different physical requirements. Since CoM sway differs between groups, this goal seems to be achieved. CoP sway was higher during the unstable foam pad conditions. Therefore, it seems to be more difficult to stabilize CoM on the foam pad. However, there was no interaction between group and condition, as expected.

The high interindividual variance of the biomechanical parameters could be explained by several factors, (1) the variety of degrees of freedom in different control processes, (2) the minimal use of resources due to the low demand of the task, (3) minimal CoM displacement not being an adequate indicator for good motor control.

According to Hodges and Tucker's model [7], the variety of degrees of freedom are reflected in the fact that different levels of the motor system can act complementarily, additively, or competitively. This implies that changes are not explained by simple, independent mechanisms like direct inhibition from nociceptive afferents on motor neurons. Aside from differences on the spinal level, motor cortex differences, which include changes in motor response planning, could be shown by other studies [19]. The relationship between the effects of pain and motor control at the spinal and cortical level are not fully understood. Though, it seems that one function of the motor system is to find a solution to protect painful structures. When pain is the highest priority process of the central nervous system it then interferes with the performance of the motor system [20].

The variety of degrees of freedom in control processes can be explained in NSLBP. For example, different pain areas or different tasks for people with NSLBP lead to different changes

**Table 1. Results for within-subjects effects.**

| Effect | Hotellings T$^2$ Value | F-Value | Hypothese df | Error df | p | η$^2$ |
|---|---|---|---|---|---|---|
| **Group** | 0.06 | 0.51 | 2 | 18 | .606 | .054 |
| **Condition** | 0.64 | 5.77 | 4 | 72 | .000 | .243 |
| **Group * Condition** | 0.02 | 0.20 | 4 | 72 | .937 | .011 |

in activity. Muscle activity could be redistributed across many trunk muscles, including across different compartments within a muscle, which influences the biomechanics of quiet standing. The measurable changes on the level of muscular activity and biomechanical outcomes in reaction to pain could thus show a high interindividual variance, but no systematic difference between groups.

A further reason for a high interindividual variance is the static nature of quiet standing, where forces acting on the trunk are low and the task is undemanding. Therefore impairments do not have a profound effect. To stabilize the postural equilibrium, different strategies can be used. One is the ankle strategy, which consists of shifting the CoM by rotating the body about the ankle like an inverted pendulum. This strategy is most common for quiet standing [21]. However, in individuals with NSLBP the other strategies seems to be impaired [22, 23]. This could explain why we did not find a difference in quiet standing.

Several new models attempt to explain postural control and emphasize using differing number of joints for their degrees of freedom. They show that models containing up to seven degrees of freedom accounted for a higher shared variance than models with fewer degrees of freedom [24]. In this case stabilizing the postural equilibrium involves a complex pattern of joint movements, where impaired hip movements may only play a minor role, especially in quiet standing. Studies [25, 26] investigating more demanding tasks like reacting to perturbations during standing did find higher CoP displacement in individuals with NSLBP.

Another factor to consider is whether optimal postural control really means that CoM displacement should be minimal. Kiemel, Zhang [27] propose that stabilizing with minimum muscle activity is the aim of postural control during quiet standing. If CoM displacements are within the range of the base of support, there is no need for the postural control system to

**Table 2. Descriptive statistics per group.**

| Parameter | Condition | Group | M | SD |
|---|---|---|---|---|
| **CoM** | **EO** | CG | 0.06 cm | 0.08 |
| | | NSLBP | 0.09 cm | 0.10 |
| | **EC** | CG | 0.06 cm | 0.08 |
| | | NSLBP | 0.09 cm | 0.10 |
| | **Foam** | CG | 0.07 cm | 0.09 |
| | | NSLBP | 0.10 cm | 0.10 |
| **EMG** | **EO** | CG | 0.5% subm. EMG | 0.2 |
| | | NSLBP | 0.6 % subm. EMG | 0.2 |
| | **EC** | CG | 1.9 % subm. EMG | 6.2 |
| | | NSLBP | 1.9 % subm. EMG | 6.1 |
| | **Foam** | CG | 1.0 % subm. EMG | 2.0 |
| | | NSLBP | 1.0 % subm. EMG | 1.9 |

EO: eyes open; EC: eyes closed; Foam: standing on a foam pad; CG: control group; NSLBP: non-specific low back pain; % subm. EMG: % of the submaximal EMG activity

**Table 3. Descriptive statistics for CoM.**

| | CoM Sway | |
| --- | --- | --- |
| | *M* | *SD* |
| **Eyes open, stable** | 0.08 cm | 0.09 |
| **Eyes closed, stable** | 0.08 cm | 0.09 |
| **Eyes open, foam** | 0.09 cm | 0.10 |

produce more muscular activity. Muscle activation would only require higher energy expenditure. Therefore, not finding significant differences could mean that due to low demands of the task CoM displacement remained within an individual's base of support in both groups. Consequently, there was also no adaptation induced by the nervous system.

Another possible reason that explains why our results are not statistically significant could be the sample size. According to the results of previous studies, we calculated the required sample size by performing a power analysis. We estimated the effect size, since it was not reported in the existing studies. With regard to publication bias [28], which is the bias against publishing non-significant results, we may have overestimated the effect size so that a larger sample size would have been necessary to find significant differences.

There is also the possibility that individuals with NSLBP can be categorized into subgroups as O'Sullivan [2] proposes. He suggests that there are three subgroups of NSLBP. In one group, movement and control impairment are secondary to pain and ultimately adaptive to pathological processes. In the second group, the dominant drive of the pain is secondary to psychological or social factors, so pain has less impact on movement. In the third group, movement and control impairments are ultimately maladaptive. Due to insufficient coping strategies a chronic abnormal tissue loading occurs. These subgroups may show different types of changes in motor control. A confirmation of differences for the entire population would thus be impossible. Alterations in motor control can further be dependent on several differentiating factors like pain intensity. There are recent reports that suggest there may be differentiating factors [11].

Nonetheless, there are some limitations to the results that also have been shown in other studies. The definition of cases is difficult since non-specific back pain is defined through excluding other diseases. Therefore, criteria like pain region and pain duration are used to determine the cases more precisely. Another problem is that studies are investigating individuals who are not the typically affected patients. For example, the average age of participants in studies is young compared to the average age of NSLBP patients receiving treatment. Since it is important to exclude spinal degeneration, the current method is to test individuals under 40 years to avoid expensive imaging methods. Recent studies did find differences between young individuals with and without NSLBP [18, 23]. Furthermore, the current pain intensity of the participants was low to moderate in the NSLBP group. The associated effects on the disability and physical scale were also low, which means that our sample was not strongly affected by NSLBP during the investigation. Though, other studies have reported differences between individuals with low pain ratings and healthy controls [29, 30].

In the future, it is necessary to identify motor control influences for specific circumstances and potentially, types of NSLBP. Furthermore, it is important to investigate tasks with a higher demand on the motor control system, such as perturbations during standing, to determine whether group differences become more evident in these situations. Additionally, taking superior parameters, like adaptability and economy of movement, into consideration would be interesting.

## Conclusion

In this study, we simultaneously recorded markers of muscle control during a quiet standing task on firm and unstable surfaces. Findings suggest that individuals with and without NSLBP may find quiet standing tasks to be undemanding. Alternatively, variation in the data for CoM sway and muscular activity of the erector spinae, could indicate that individuals diagnosed with NSLBP could fall into different categories. Further study is warranted to more closely examine instantaneous control processes across a variety of physical tasks for individuals with NSLBP to better understand implications of low back pain in movement and potential approaches of treatment.

## Author Contributions

**Conceptualization:** Cathrin Koch, Stefan Göbel, Frank Hänsel.

**Data curation:** Cathrin Koch, Augusto Garcia-Augundez.

**Formal analysis:** Cathrin Koch, Augusto Garcia-Augundez.

**Funding acquisition:** Cathrin Koch, Stefan Göbel, Frank Hänsel.

**Investigation:** Cathrin Koch.

**Methodology:** Cathrin Koch, Augusto Garcia-Augundez, Frank Hänsel.

**Project administration:** Cathrin Koch, Stefan Göbel, Frank Hänsel.

**Resources:** Stefan Göbel, Frank Hänsel.

**Software:** Augusto Garcia-Augundez.

**Supervision:** Stefan Göbel, Frank Hänsel.

**Validation:** Cathrin Koch, Augusto Garcia-Augundez.

**Visualization:** Cathrin Koch, Augusto Garcia-Augundez.

**Writing – original draft:** Cathrin Koch, Frank Hänsel.

**Writing – review & editing:** Augusto Garcia-Augundez, Frank Hänsel.

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
