## [Decision Letter · Decision Letter 0]

25 Mar 2020

PONE-D-19-34384

A case control study to investigate differences in motor control between individuals with and without non-specific low back pain during standing

PLOS ONE

Dear Prof. Dr. rer. nat. Hänsel,

Thank you for submitting your manuscript to PLOS ONE. After careful consideration, we feel that it has merit but does not fully meet PLOS ONE’s publication criteria as it currently stands. Therefore, we invite you to submit a revised version of the manuscript that addresses the points raised during the review process.

We would appreciate receiving your revised manuscript by May 09 2020 11:59PM. To enhance the reproducibility of your results, we recommend that if applicable you deposit your laboratory protocols in protocols.io, where a protocol can be assigned its own identifier (DOI) such that it can be cited independently in the future. For instructions see: http://journals.plos.org/plosone/s/submission-guidelines#loc-laboratory-protocols

We look forward to receiving your revised manuscript.

Kind regards,

Giulia Bivona

Academic Editor

PLOS ONE

Journal Requirements:

2. Please include additional information regarding the survey or questionnaire used in the study and ensure that you have provided sufficient details that others could replicate the analyses. For instance, if you developed a questionnaire as part of this study and it is not under a copyright more restrictive than CC-BY, please include a copy, in both the original language and English, as Supporting Information. In addition, please refrain from reporting p values as .000, either report the exact value or use the format p<0.0001.

Reviewers' comments:

Reviewer's Responses to Questions

**Comments to the Author**

1. Is the manuscript technically sound, and do the data support the conclusions?

Reviewer #1: Yes

Reviewer #2: Yes

2. Has the statistical analysis been performed appropriately and rigorously? 

Reviewer #1: Yes

Reviewer #2: Yes

3. Have the authors made all data underlying the findings in their manuscript fully available?

Reviewer #1: Yes

Reviewer #2: Yes

4. Is the manuscript presented in an intelligible fashion and written in standard English?

Reviewer #1: Yes

Reviewer #2: No

5. Review Comments to the Author

Reviewer #1: I really appreciate the study for its Innovative correlation pattern. I would like to recommend using the visual analogue scale analysis in the future similar trials so that a quantitative estimation of the NSLBP can enrich the correlation.

Reviewer #2: This manuscript documents a study comparing those with non-specific low back pain (NSLBP) to age- and sex-matched healthy control participants in terms of trunk muscle activity and centre of pressure control during three quiet standing conditions. The authors were unable to determine any group differences but did show that one of their measures was affected by standing condition despite showing some observable differences in descriptive statistics.

I believe there is scientific merit in this manuscript, but the writing and presentation needs substantial improvement to improve the quality and clarity of this work. The majority of my comments reflect minor technical additions/omissions, however I still suggest a major revision to emphasize the amount of work needed to improve the language.

General Comments

The manuscript is filled with grammatical and phrasing errors that make the manuscript difficult to read and understand. I recommend the authors have a fluent English speaker completely review the grammar and phrasing. Examples of particularly difficult to parse passages include, but are not limited to, lines 54-55, 62-68, 74-76, 84-86, 132, 219-220, 305-309.

The authors did a good job documenting their NSLBP group, but it appears to be heterogeneous. Most of the prior work the authors cited in their introduction focused on specific subgroups of the NSLBP population. An advantage to this technique is that those other authors can focus on a specific aspect of low back pain development. The purpose of this papers states that the authors wanted to focus on altered motor control, and their standing conditions appear to reflect that. However, their NSLBP population may have been too generic to truly suss out mechanisms of low back pain development related to altered motor control despite controlling for . Subgroups including the Stabilization/Recurrent group in the Treatment-based Classification, parts of the Movement System Impairment paradigm, or the updated O’Sullivan classification scheme (what the authors cited in the discussion in lines 291-300) are more specific and more likely to show a common deviation with respect to a control group. Can the authors rationalize the lack of subgrouping their NSLBP group?

The use of RMS to examine centre of pressure data is common, however it is more sensitive high-frequency changes than a linear measure such as the average or mean absolute value. The authors did not filter their force plate data and collected the data at a high sample rate relative to the frequency content of their signal (quiet standing). This may have washed out potential true differences between their NSLBP and control groups, with the effects more exaggerated the further participants stood away from the force plate origin. I suggest the authors use a metric less sensitive to high-frequency noise (such as the mean absolute value) or remove the higher frequency content prior to computing the RMS on their centre of pressure data. The authors should also report where participants stood relative to the force plate origin.

Specific Comments

Line 58 – the differences in trunk muscle activity and centre of pressure findings precede discomfort development in this group

Line 69 – The paragraph starting here seems unnecessarily long to convey the motivation behind use of centre of pressure or EMG data

Line 128 – p’s ≥ 0.210?

Line 137 – Are these tasks too easy to differentiate control and NSLBP groups?

Line 143 – More detail is required about the mood state questionnaire.

Line 144 – What was the maximum rating of the numeric rating scale?

Line 150 – What were the kinematic data collected? Does this refer to the force plate data? If so, “kinematic” is not required, otherwise, please describe the other kinematic data.

Line 175 – How did the authors remove heart rate contamination from their EMG data? It is also unclear when this occurred in the data analysis relative to the subMVC normalization.

Line 178 – The statistical modelling description seems out of place in the Results section, consider moving back to the Methods.

Lines 188 to 192 – Which numbers correspond to the control group, and which numbers match the NSLBP group?

Line 195 – I believe group should be a between-subjects factor, not a repeated factor

Line 228 – Can the authors report an effect size for the non-significant between-group difference in centre of pressure data?

Line 229 – Why is higher centre of mass sway bad? Were participants instructed to stand as still as possible?

Line 239 – This concluding sentence requires more information.

Lines 258 to 268 – This paragraph is seems distracted and does not add much to the overall discussion as is.

6. PLOS authors have the option to publish the peer review history of their article (what does this mean?). If published, this will include your full peer review and any attached files.

Reviewer #1: Yes: MM

Reviewer #2: No

---

## [Author Response · Author response to Decision Letter 0]

27 May 2020

Reviewer #1: 

I really appreciate the study for its Innovative correlation pattern. I would like to recommend using the visual analogue scale analysis in the future similar trials so that a quantitative estimation of the NSLBP can enrich the correlation.

Response: 

Thank you for your suggestion. We have considered the use of the VAS. To avoid overburdening participants we decided to use the numeric rating scale. Nonetheless your argument for the VAS is convincing. For the future, we plan on using the VAS to enrich the correlation. 

Reviewer #2: 

This manuscript documents a study comparing those with non-specific low back pain (NSLBP) to age- and sex-matched healthy control participants in terms of trunk muscle activity and centre of pressure control during three quiet standing conditions. The authors were unable to determine any group differences but did show that one of their measures was affected by standing condition despite showing some observable differences in descriptive statistics.

I believe there is scientific merit in this manuscript, but the writing and presentation needs substantial improvement to improve the quality and clarity of this work. The majority of my comments reflect minor technical additions/omissions, however I still suggest a major revision to emphasize the amount of work needed to improve the language.

Response: 

Thank you for your suggestions on language improvement. The manuscript has been reviewed by a native English speaker.

General Comments 

The manuscript is filled with grammatical and phrasing errors that make the manuscript difficult to read and understand. I recommend the authors have a fluent English speaker completely review the grammar and phrasing. Examples of particularly difficult to parse passages include, but are not limited to, lines 54-55, 62-68, 74-76, 84-86, 132, 219-220, 305-309.

Response: 

The manuscript has been reviewed by a native English speaker.

The authors did a good job documenting their NSLBP group, but it appears to be heterogeneous. Most of the prior work the authors cited in their introduction focused on specific subgroups of the NSLBP population. An advantage to this technique is that those other authors can focus on a specific aspect of low back pain development. The purpose of this papers states that the authors wanted to focus on altered motor control, and their standing conditions appear to reflect that. However, their NSLBP population may have been too generic to truly suss out mechanisms of low back pain development related to altered motor control despite controlling for . Subgroups including the Stabilization/Recurrent group in the Treatment-based Classification, parts of the Movement System Impairment paradigm, or the updated O’Sullivan classification scheme (what the authors cited in the discussion in lines 291-300) are more specific and more likely to show a common deviation with respect to a control group. Can the authors rationalize the lack of subgrouping their NSLBP group?

Response: 

Prior to this case-control study we conducted a review on motor control in individuals with LBP. In the current study our goal was to replicate those findings with a high quality study. The studies included in our Review classify non-specific back pain according to the duration of pain, pain area, or degree of severity. Based on the common classifications in these studies we defined these criteria precisely to generate a homogenous group.

We think it is a valuable comment to consider subgroups as we have now pointed out in our discussion. A further validation here is necessary, but it would be interesting to incorporate subgroups, like for example O´Sullivan suggests, in our further research. 

The use of RMS to examine centre of pressure data is common, however it is more sensitive high-frequency changes than a linear measure such as the average or mean absolute value. The authors did not filter their force plate data and collected the data at a high sample rate relative to the frequency content of their signal (quiet standing). This may have washed out potential true differences between their NSLBP and control groups, with the effects more exaggerated the further participants stood away from the force plate origin. I suggest the authors use a metric less sensitive to high-frequency noise (such as the mean absolute value) or remove the higher frequency content prior to computing the RMS on their centre of pressure data. The authors should also report where participants stood relative to the force plate origin.

Response: 

Standing position includes that participants were standing in the middle of force plate with their medial malleolus at the center with the same distance for the right and left foot. We have added this detail to our manuscript.

Correctly, you mention the problems that occur with the use of the RMS, but since our goal was to make the results of our study comparable to those of other studies, we decided to use RMS, which is the method in the vast majority of studies. 

Additionally, we have now applied a 5Hz low pass filter. 

Specific Comments

Line 58 – the differences in trunk muscle activity and centre of pressure findings precede discomfort development in this Group

Response: 

We have made changes here to address this comment.

Line 69 – The paragraph starting here seems unnecessarily long to convey the motivation behind use of centre of pressure or EMG data

Response: 

We have shortened this paragraph.

Line 128 – p’s ≥ 0.210?

Response: 

We have added an apostrophe.

Line 137 – Are these tasks too easy to differentiate control and NSLBP groups?

Response: 

This is a good point that we have considered. We mention it in the discussion now. Nonetheless our aim was to investigate quiet standing as specific subgroup of standing and there are previous studies that have found differences in quiet standing as our review shows. 

Line 143 – More detail is required about the mood state questionnaire.

Response: 

We have specified the questionnaire. It’s only purpose was to document stress during the tests, so that we could be assured that the stress itself of the test was not having any impact on the participants, or results. 

Line 144 – What was the maximum rating of the numeric rating scale?

Response: 

0-10; we added this into the description.

Line 150 – What were the kinematic data collected? Does this refer to the force plate data? If so, “kinematic” is not required, otherwise, please describe the other kinematic data.

Response: 

Yes, it is only force plate data. We have changed the term.

Line 175 – How did the authors remove heart rate contamination from their EMG data? It is also unclear when this occurred in the data analysis relative to the subMVC normalization.

Response: 

We added the concrete bandpass filter.

Line 178 – The statistical modelling description seems out of place in the Results section, consider moving back to the Methods.

Response: 

We have discussed and understand your point. We decided to remove this sentence completely since it is common practice. We also report the p-values. 

Lines 188 to 192 – Which numbers correspond to the control group, and which numbers match the NSLBP group?

Response: 

We have clarified that in this section.

Line 195 – I believe group should be a between-subjects factor, not a repeated factor

Response: 

Since we matched the groups we can treat them as paired groups. Therefore, we have kept it as a repeated factor. We now point out more clearly that the groups were matched.

Line 228 – Can the authors report an effect size for the non-significant between-group difference in centre of pressure data?

Response: 

We added an effect size for the non-significant between group difference in center of pressure.

Line 229 – Why is higher centre of mass sway bad? Were participants instructed to stand as still as possible?

Response: 

Yes, participants were instructed to stand as still as possible. We added this in the methods section

Line 239 – This concluding sentence requires more information.

Response: 

We provided more information to make the sentence clearer.

Lines 258 to 268 – This paragraph is seems distracted and does not add much to the overall discussion as is.

Response: 

We recognize that our argument might not be clear enough. In this paragraph we now introduce the idea that different balance strategies depend on the demand of the task, and emphasize that for low demand tasks, minimal use of resources can lead to non-significant results. From our perspective, this is an important point to better classify our results. Therefore, we have shortened the paragraph and clarified our argument.

---

## [Editor Report · Decision Letter 1]

4 Jun 2020

A case control study to investigate differences in motor control between individuals with and without non-specific low back pain during standing

PONE-D-19-34384R1

Dear Dr. Hänsel,

We’re pleased to inform you that your manuscript has been judged scientifically suitable for publication and will be formally accepted for publication once it meets all outstanding technical requirements.

Kind regards,

Giulia Bivona

Academic Editor

PLOS ONE
---

## [Editor Report · Acceptance letter]

9 Jun 2020

PONE-D-19-34384R1 

A case control study to investigate differences in motor control between individuals with and without non-specific low back pain during standing 

Dear Dr. Hänsel:

I'm pleased to inform you that your manuscript has been deemed suitable for publication in PLOS ONE. Congratulations! Your manuscript is now with our production department. 

Kind regards, 

on behalf of

Dr. Giulia Bivona 

Academic Editor

PLOS ONE